# OpenReview forum: "TAN: Topological Attention Networks for Hierarchical Document Understanding"
_ICLR.cc/2026/Conference — ICLR 2026 Conference Withdrawn Submission_

### Official Review · Reviewer_dijH · 2025-10-16

**Soundness:** 1
**Presentation:** 3
**Contribution:** 2
**Rating:** 2
**Confidence:** 5

**Summary:**

The paper introduces Topological Attention Networks (TAN), a Transformer architecture that incorporates topological data analysis (TDA) into the attention mechanism. TAN uses a topology-guided locality-sensitive hashing (LSH) to reduce the quadratic complexity of self-attention while preserving structural relationships in text. It constructs k-nearest neighbor graphs in embedding space and aggregates multi-scale features through learnable topological encoders, enabling efficient processing of long and hierarchical documents.

**Strengths:**

This paper presents an interesting approach addressing an important issue: optimizing the quadratic complexity of Transformer-based methods, and theoretically validates the feasibility of the approach.

**Weaknesses:**

1. The main issue with this paper is the insufficiency of the experiments. The primary goal of optimizing Transformers is to reduce the computational cost of modeling long contexts, yet the authors only evaluate their method on two relatively short datasets in the experimental section.
2. Although several models are compared, these baselines are still outdated and do not demonstrate the advantage of TAN over more recent methods. A very natural question is how TAN performs compared to Hyena or other more recent models. Moreover, the authors claim that sparse attention patterns reduce complexity but sacrifice global context, while low-rank approximations lose fine-grained details critical for hierarchical understanding. Therefore, TAN should be thoroughly compared with these representative methods to validate its claimed superiority.
3. TAN lacks comparisons with large language models (LLMs). While I do not claim that such a comparison would be decisive, given the remarkable ability of many existing LLMs to handle long contexts, a natural question arises: what advantage does TAN offer over MoE-based LLMs—is it in performance, computational efficiency, or both?
4. Additionally, considering that TAN modifies the fundamental Transformer architecture, the performance of standard Transformers on some NLP benchmarks should be further reported to ensure the generalizability of TAN.
5. From Table 3, TAN appears to be suboptimal, showing no clear advantage and even performing much worse than the TFIDF+SVM method. The authors suggest this may be due to training resource limitations, but training 40,000 samples for 10 or 20 epochs would not incur unacceptable costs. Why did the authors not conduct a fair comparison?
6. In addition, some formatting and presentation details of the paper still need improvement. For example, Figure 1 contains obvious missing or incorrect symbols, and some key experiments, such as ablation studies, are placed in the appendix.

**Questions:**

See Weaknesses.

---

### Official Review · Reviewer_Yz7X · 2025-10-25

**Soundness:** 1
**Presentation:** 2
**Contribution:** 2
**Rating:** 2
**Confidence:** 2

**Summary:**

This paper proposes a modification to standard self-attention that integrates topological features into the attention mechanism. To improve efficiency, the proposed approach employs locality-sensitive hashing guided by these topological features. The mechanism groups tokens into buckets based on their topological similarity, allowing attention computations to be performed only within each bucket. The main innovation lies in incorporating topological features into the hash function.

The authors evaluate their method on two datasets: GoEmotions (multi-label classification) and LEDGAR (multi-class classification).

**Strengths:**

The paper is easy to follow and the main contribution (modification) is clearly explained.

**Weaknesses:**

I feel there is a disconnection between the authors' motivation and their empirical experiments.

* The authors motivate their approach by modeling structural patterns in documents. However, they only use two benchmark datasets which contain sentences (GoEmotations, 20 tokens) and paragraphs (LEDGAR, 150 tokens). I suggest that the authors use real document classification tasks involving documents longer than 512 or even 4,096 tokens.
* The main advantage of the proposed approach should be its efficiency compared to standard self-attention. However, there is no empirical comparison to support this claim, except for computational complexity comparision.

The chosen baselines may be outdated or relatively weak. Even though, the proposed method performs significantly worse than all baselines—including TFIDF-SVM—on LEDGAR. The authors should use empirical evidence to support the claim "TAN would demonstrate even stronger performance advantages with larger-scale datasets and extended training, as the topological inductive bias should become more pronounced with increased data diversity and training duration"

**Questions:**

* what is $T$ in Eq6? is it the same as $T_i$ in Eq 2 and $T(x)$ in Eq8?

---

### Official Review · Reviewer_S5P2 · 2025-10-29

**Soundness:** 1
**Presentation:** 1
**Contribution:** 1
**Rating:** 0
**Confidence:** 4

**Summary:**

This paper introduces Topological Attention Networks (TAN), a novel architecture that integrates topological features into the attention mechanism to achieve sub-quadratic complexity while maintaining competitive performance on text classification tasks. The authors evaluate TAN on GoEmotions and LEDGAR datasets, demonstrating improvements over baseline models.

**Strengths:**

- The paper contains original idea (probably? hard to say actually due to very unclear presentation)

**Weaknesses:**

The paper contains blatant AI slop and has low quality overall:

- Figure 1 in the main part of the paper doesn't make sense and seem to be AI-generated. For example, what does the formula "$MLP = \frac{1}{G}\frac{\Lambda}{ }$" mean - in the "Gating **Mechenism**" block (the authors' spelling)? What does the formula "$G*C + (1-)H_{in}$" mean in the "Integration" block? These formulas are clearly meaningless.
- The model description in the main text is very unclear and contradicts Appendix A.
- The authors do NOT provide a mathematical proof of their "subquadratic complexity" claim, nor do they provide experimental validation of this claim. No actual runtime comparisons or memory usage statistics are provided.
- The paper claims scalability and usefulness for long documents and "hierarchical" structures, yet its empirical evidence comes almost entirely from short or medium-length inputs (GoEmotions contains at average ≈20 tokens and LEDGAR contains at average ≈150 tokens, according to authors themselves), leaving the long-context claim unsubstantiated.
- The baseline results on GoEmotions dataset appear unusually low - for example, for RoBERTa-large model. These numbers contradict typical performance expectations and raise questions about implementation correctness.

**Questions:**

On Figure 1:

What is Sigiod(o)?

**Details Of Ethics Concerns:**

Figure 1 in the main part of the paper doesn't make sense and seem to be AI-generated. This is unacceptable, because, by the logic of the paper, Figure 1 should depict the proposed model’s architecture and workflow, not a generic AI-generated image.

---

### Note · Authors · 2025-12-03

I have read and agree with the venue's withdrawal policy on behalf of myself and my co-authors.